# Molecular Analysis of SARS-CoV-2 Spike Protein-Induced Endothelial Cell Permeability and vWF Secretion

**DOI:** 10.3390/ijms24065664

**Published:** 2023-03-16

**Authors:** Yuexi Guo, Venkateswarlu Kanamarlapudi

**Affiliations:** Institute of Life Science, School of Medicine, Swansea University, Singleton Park, Swansea SA2 8PP, UK

**Keywords:** COVID-19, SARS-CoV-2, spike protein, RBD, vWF, endothelial permeability, ACE2, ARF6, signaling

## Abstract

Coronavirus disease COVID-19, which is caused by severe acute respiratory syndrome coronavirus SARS-CoV-2, has become a worldwide pandemic in recent years. In addition to being a respiratory disease, COVID-19 is a ‘vascular disease’ since it causes a leaky vascular barrier and increases blood clotting by elevating von Willebrand factor (vWF) levels in the blood. In this study, we analyzed in vitro how the SARS-CoV-2 spike protein S1 induces endothelial cell (EC) permeability and its vWF secretion, and the underlying molecular mechanism for it. We showed that the SARS-CoV-2 spike protein S1 receptor-binding domain (RBD) is sufficient to induce endothelial permeability and vWF-secretion through the angiotensin-converting enzyme (ACE)2 in an ADP-ribosylation factor (ARF)6 activation-dependent manner. However, the mutants, including those in South African and South Californian variants of SARS-CoV-2, in the spike protein did not affect its induced EC permeability and vWF secretion. In addition, we have identified a signaling cascade downstream of ACE2 for the SARS-CoV-2 spike protein-induced EC permeability and its vWF secretion by using pharmacological inhibitors. The knowledge gained from this study could be useful in developing novel drugs or repurposing existing drugs for treating infections of SARS-CoV-2, particularly those strains that respond poorly to the existing vaccines.

## 1. Introduction

Coronavirus disease (COVID-19) is an infectious disease that has recently become a threat to human health and life globally. The World Health Organization officially declared the outbreak as a public health emergency of international concern on 30 January 2020 [1]. It is caused by severe acute respiratory syndrome coronavirus (SARS-CoV-2). The genetic sequence of the novel coronavirus suggests that it is similar to the coronavirus found in bats but differs from other coronaviruses such as the SARS coronavirus and the Middle East Respiratory Syndrome coronavirus (MERS) [2]. In contrast to SARS and MERS, SARS-CoV-2 causes severe acute respiratory illness and is highly contagious. SARS-CoV-2 affects not only the respiratory tract but also the heart, gastrointestinal system, liver, kidney, and central nervous system, ultimately leading to multi-organ failure [2,3]. SARS-CoV-2 enters the host human cells by using the angiotensin-converting enzyme (ACE)2 receptor. The spike protein of SARS-CoV-2, which plays a key role in the receptor recognition and cell membrane fusion process, is composed of two subunits, S1 and S2. The S1 subunit contains a receptor-binding domain (RBD) that recognizes and binds to the host receptor ACE2 whereas the S2 subunit mediates viral and host cell membrane fusion [4]. It has been suggested that blocking the entry of the virus by inhibiting ACE2 is more advantageous than inhibiting the subsequent stages of the SARS-CoV-2 life cycle [5]. Since its spike protein plays an important role in viral infection, it has become an important target for vaccine development, antibody-blocking therapy, and small molecule inhibitors [4].

The severity of COVID-19 is closely related to the presence of the so-called “cytokine storm” (the increased cytokine levels), which is an exaggerated immune response by the patients to SARS-CoV-2 infection. The patients show the elevation of proinflammatory cytokines such as IL-1β, IL-6, and TNF-α levels in the blood [6,7]. However, cytokine storms are also observed in non-COVID acute respiratory syndromes [8]. SARS-CoV-2 activates NF-κB and inhibition of the NF-κB pathway enhances the survival rate of mice infected with SARS-CoV-2 [9]. Therefore, these studies have suggested that the activation of the nuclear factor NF-κB signaling pathway represents a major contributor to cytokine storm-induced inflammation after SARS-CoV-2 infection [7]. In addition, ADP-ribosylation factor (ARF)6, which is a Ras-related small GTPase and plays an important role in cancer [10], regulates the entry of the SARS-CoV-2 virus into the host cells [11]. The similarities between COVID-19 and bacterial lipopolysaccharide (LPS)-induced sepsis have been identified by comparing the immunopathogenesis and pathophysiology of these two infections [12]. Moreover, the ARF6 activation leads to an unstable and leaky vasculature in LPS-induced sepsis. These studies suggest that ARF6 could be a potential target for COVID-19.

It is currently believed that, in addition to being a respiratory disease, COVID-19 might also be a ‘vascular disease’ [13,14,15], as it may result in a leaky vascular barrier and increased expression of von Willebrand factor (vWF), which is responsible for increased coagulation, cytokine release, and inflammation [16,17,18,19,20,21,22,23,24,25]. In addition, thrombotic events occur in COVID-19 patients, which is known as COVID-19-associated coagulopathy (CAC) and endothelial-derived vWF may play a significant role in it. COVID-19 seemingly affects several vital organs through vascular dysfunction by altering endothelial cell (EC) permeability [26]. Furthermore, the main cause of mortality in COVID-19 patients is microvascular thrombosis [27]. Other studies have indicated that the SARS-CoV-2 spike protein disrupts the integrity of the endothelial barrier [28,29]. When 26 out of 29 proteins encoded by the SARS-CoV-2 genome were expressed individually in endothelial cells, most proteins altered the endothelial cell (EC) permeability and increased vWF expression [30]. This indicates that vWF may serve as a potential target in COVID-19 pathogenesis. SARS-CoV-2 has also been shown to induce reactive oxygen species (ROS) production and promote oxidative stress, for instance, through TNF-α and NF-κB signaling pathways. Since the mitochondria are vital for cellular oxidative homeostasis, the heightened inflammatory/oxidative state may cause mitochondrial dysfunction, resulting in cell apoptosis [31]. 

Thrombin cleavage of PAR1 results in phosphorylation of myosin light chain (MLC), leading to the reorganization of the actin cytoskeleton into contractile stress fibers and opening of intercellular gaps, causing the increase of EC permeability [32,33,34]. It is known that Rho-associated kinases 1 and 2 (ROCK1/2) are Rho-GTPase effectors that control key aspects of the actin cytoskeleton [35], MLC, which is phosphorylated (MLC-p) and activated by ROCK [36]. However, how COVID-19 alters endothelial barrier function has not been understood very well. In this study, we established that the SARS-CoV-2 spike protein S1 RBD is sufficient to induce EC permeability and its vWF secretion. The RBD-induced EC permeability and vWF secretion are unaltered with various mutations, including those found in South African (SA) and South Californian (SC) variants, in the spike protein. We also showed that the SARS-CoV-2 spike protein S1 RBD increases the ACE2 expression and ARF6 activation in endothelial cells. Additionally, we assessed the effects of the inhibitors of various signaling molecules described above, which are associated directly or indirectly with COVID-19, on inhibiting SARS-CoV-2 spike protein-induced EC permeability and vWF secretion.

## 2. Results

### 2.1. SARS-CoV-2 Spike Protein S1 RBD Induces EC Permeability and vWF Secretion

To study how SARS-CoV-2 infection impacts leaky endothelial vascular permeability, we assessed in vitro the dose- and time-dependent effect of the SARS-CoV-2 spike protein S1 RBD on the permeability of EA.hy 296 cells, which are widely used for endothelial cell (EC) permeability assays [37,38,39]. For this purpose, we used a novel trans-monolayer permeability assay, which is useful for the visualization and rapid quantification of EC permeability by fluorescence microscopy [40]. The basis of this assay is FITC-conjugated streptavidin passing through the intercellular gap in the cell monolayer and thereby binding to biotinylated gelatin located underneath the cell monolayer. No cells and untreated cells were used as positive (100% permeability) and negative, respectively, controls in each assay. When EA. hy296 cells were treated without or with 1 µg/mL of SARS-CoV-2 spike S1 RBD for 24–120 h, we saw a time-dependent increase, particularly from 72 h of treatment onwards, in the EC permeability both in the RBD-treated and -untreated cells (Figure 1a). However, the EC permeability was much higher in the RBD-treated cells, reaching approximately 60% of the maximum permeability after 120 h of treatment (Figure 1a). In the RBD-untreated cells, the permeability was less than 20% at the same time point. When we stained the RBD-untreated and -treated cells with DAPI (nuclear staining), we saw no change in the DAPI staining intensity between the samples (Appendix A). This indicates that the EC permeability increase in the RBD-treated cells is not due to the loss of cells upon the RBD treatment. The cells were viable only for 120 h and, therefore, we did not analyze the RBD effect on the EC permeability beyond the 120 h of incubation. 

Since vWF levels also increase in COVID-19 patient blood and endothelial cells are known to secrete vWF [22,41], we also analyzed the time-dependent effect of the RBD treatment on vWF secretion from the EA.hy 296 cell monolayer. This was performed by estimating vWF levels by ELISA in the media of cells treated without or with the RBD for 24–120 h (Figure 1b). vWF secretion increased significantly in a time-dependent manner by 24 h of the treatment only in the RBD-treated cells and reached the maximum levels after 120 h of the treatment. Because COVID-19 causes a cytokine storm by increasing pro-inflammatory cytokines such as TNF-α, IL-6, and IL-1β, we also analyzed these cytokine levels in the media of cells treated without or with the RBD for 24–120 h (Appendix A). However, there was only a negligible increase or no increase in their secretion from the cells treated with the RBD for 120 h.

Since we have seen a time-dependent increase of EC permeability by the SARS-CoV-2 spike S1 RBD, we also assessed the dose-dependent effect of the SARS-CoV-2 spike protein S1 RBD on the permeability of EA.hy 296 cells. For this purpose, cells were treated with 0–10 µg/mL of the SARS-CoV-2 spike protein S1 RBD for 72 h. The EC permeability was increased in a dose-dependent manner from 0.01–10 µg/mL and reached ~50% in the 10 µg/mL RBD-treated cells (Figure 2a). We did not see any difference in nuclear (DAPI) staining of cells treated without or with 0.01–10 µg/mL S1 RBD (Appendix A), indicating no loss of cells when treated with various doses of the S1 RBD. Subsequently, the dose-dependent effect of the RBD treatment on vWF secretion from EA.hy 296 cell monolayer was evaluated by measuring vWF levels by ELISA in the media of cells treated with 0–10 µg/mL RBD for 72 h. vWF secretion was also increased significantly in a dose-dependent manner from 0.01 µg/mL of RBD-treated cells and reached the maximum levels in 10 µg/mL of RBD-treated cells (Figure 2b). However, we did not see any noticeable increase in the secretion of pro-inflammatory cytokines (TNF-α, IL-6, and IL-1β) from the cells treated with 0–10 µg/mL RBD for 72 h (Appendix A). Unless otherwise indicated, a standard treatment of 1 µg/mL of the SARS-CoV-2 spike S1 RBD for 72 h was used in further experiments. This is because it not only produced a clear response but would also allow an analysis of the positive or negative effects of the treatments on the RBD-induced permeability in further experiments.

### 2.2. The Mutations of SARS-CoV-2 Spike Protein S1 Unaffected Its Induced EC Permeability and vWF Secretion 

It is well known that the SARS-CoV-2 virus spike protein S1 mutates over time, which leads to genetic variation in the circulating viral strains that seriously impact human health [42]. So, we next determined whether mutations in the SARS-CoV-2 spike protein have any effect on the spike protein S1-induced EC permeability and vWF secretion. For this, first, we analyzed the time-dependent effect of the SARS-CoV-2 spike protein S1 on the permeability of EA.hy 296 cells. When the cells were treated with 2.5 µg/mL of the SARS-CoV-2 spike protein S1 (since it is approximately thrice as large as the RBD) for 0–120 h and without for 120 h, we observed a time-dependent increase in the permeability, particularly from 96 h of treatment onwards (Figure 3a). The EC permeability reached the maximum (~45%) after 120 h of the treatment, which is much higher than that observed in the spike protein S1-untreated cells for 120 h as expected. Likewise, vWF secretion increased significantly from 24 h of the treatment only in the SARS-CoV-2 spike protein S1-treated cells and reached the maximum levels after 120 h of the treatment (Figure 3b). After establishing the optimal time (96 h) to induce the EC permeability by the spike protein S1, we analyzed the relative effect of various SARS-CoV-2 spike protein S1 mutants on the permeability of EA.hy 296 cells and its vWF secretion. When EA.hy 296 cells were treated with 2.5 µg/mL of SARS-CoV-2 (WT) or its mutants for 96 h, we found a significant increase in the EC permeability in all the treatments (Figure 3c). However, no significant difference was observed between the WT and mutants in inducing the EC permeability. Similarly, no detectable difference was seen between the WT- and mutants-induced vWF secretion from EA.hy 296 cells (Figure 3d). Taken together, these results suggest that the SARS-CoV-2 spike protein S1-induced EC permeability and vWF secretion are unaffected by the mutations in the spike protein that we analyzed.

### 2.3. ACE2 Expression and ARF6 Activation Are Increased in EA.hy 296 Cells Treated with SARS-CoV-2 Spike Protein S1 RBD

Since SARS-CoV-2 requires ACE2 and ARF6 to enter human cells [43,44], we assessed where ACE2 expression and ARF6 activation are altered in SARS-CoV-2 spike S1 RBD- stimulated EA.hy 296 Cells. For this purpose, we lysed EA.hy 296 cells treated with 0–10 μg/mL of the RBD for 72 h and used the cell lysates for analyzing the expression of ACE2 expression and ARF6 activation. When the ACE2 expression was analyzed by immunoblotting using the ACE2 inhibitory antibody, we saw a slight increase in ACE2 protein expression in the 0.01 μg/mL RBD-treated cells but the expression did not increase further as the RBD amount increased in the treatment (Figure 4a). 

The ARF family of small GTPases cycles between GDP-bound inactive and GTP-bound active conformations. Guanine nucleotide exchange factors (GEFs) activate ARFs whereas GTPase activating proteins (GAPs) inactive them (Figure 4b) [45]. Since ARF6 is considered a potential target in SARS-CoV-2 infection [44], we used the GST-GGA3 PBD pulldown assay to assess if the SARS-CoV-2 spike protein S1 RBD treatment of EA.hy 296 cells causes ARF6 activation. The GST-GGA3 PBD specifically binds to the GTP-bound form of ARF and, therefore, the fusion protein coupled to glutathione beads along with an ARF1- or ARF6-specific antibody has been used to detect activated ARF1 or ARF6 in cell lysates [46]. Upon treatment with the RBD, the levels of ARF6-GTP, but not ARF1-GTP, were markedly increased in EA.hy 296 cells (Figure 4c,d), indicating that the SARS-CoV-2 spike S1 RBD selectively activates ARF6 in the cells. 

### 2.4. The EC Permeability and vWF Secretion in SARS-CoV-2 Spike Protein S1 RBD-Stimulated EA.hy 296 Cells Is Mediated by ACE2

As we showed that the SARS-CoV-2 Spike protein S1 RBD upregulates ACE2 expression and ARF6 activation, we next determined the effect of the activation of ACE2 and ARF6 on the RBD-induced EC permeability and vWF secretion. Several studies reported a promising anti-COVID-19 drug candidate, the lipoglycopeptide antibiotic dalbavancin, which directly binds to ACE2 with high affinity and thereby blocks its interaction with the SARS-CoV-2 spike protein in vitro [47,48]. The ACE2 inhibitory antibody is also shown to bind ACE2 and thereby inhibit SARS-CoV-2 virus entry into the cells [49,50]. Therefore, we hypothesized that dalbavancin and the ACE2 inhibitory antibody may affect SARS-CoV-2 spike protein-induced EC permeability and vWF secretion in EA.hy 296 cells. For this purpose, dalbavancin or the ACE2 inhibitory antibody pretreated EA.hy 296 cells were used to assess RBD-induced EC permeability and vWF secretion. We observed that dalbavancin and the ACE2 inhibitory antibody markedly decreased the EC permeability (Figure 5a) and vWF secretion (Figure 5b) in the RBD-treated cells. However, the solvent DMSO or control goat IgG did not affect the RBD-induced EC permeability and vWF secretion. 

### 2.5. Dissecting the Signaling Pathway for EC Permeability and vWF Secretion in SARS-CoV-2 Spike Protein S1 RBD-Stimulated EA.hy 296 Cells 

We next used the pharmacological inhibitors of ARF6 activation (chlortetracycline) and inactivation (QS11), NF-kB (SC-514), reactive oxygen species (ROS) (N-Acetylcysteine [NAC]), and ROCK1/2 (Y27632) to define the signaling cascade downstream of ACE2 in the EC permeability and vWF secretion in the SARS-CoV-2 spike S1 RBD-stimulated EA.hy 296 cells (Figure 6a). In this study, we determined that the SARS-CoV-2 spike S1 RBD increases ARF6 activation in EA.hy 296. Chlortetracycline is a novel inhibitor for ARF6 activation whereas QS11 is a broad-spectrum ARF GAP inhibitor that prevents ARF6 inactivation [51,52]. Studies have found that SARS-CoV-2 activates NF-κB in a dose-dependent manner, and inhibition of the NF-κB pathway enhances survival rates in mice infected with SARS-CoV-2 [53]. NAC inhibits oxidative stress by reducing mitochondrial dysfunction [28]. Y27632 is a competitive inhibitor of both ROCK1 and ROCK2, which play an important role in Thrombin-induced endothelial permeability [35,54]. 

We hypothesized that these inhibitors may affect the RBD-induced EC permeability and vWF secretion by inhibiting the signaling pathway downstream of ACE2. To test this hypothesis, we analyzed the effect of chlortetracycline, QS11, SC-514, NAC, and Y27632 pretreatment on the EC permeability and vWF secretion in RBD-treated EA.hy 296 cells. We found that Chlortetracycline, SC-514, NAC, and Y27632 significantly inhibited the RBD-induced EC permeability (Figure 6b) as well as vWF secretion (Figure 6c), whereas they did not show any effect on the RBD-untreated cells.

### 2.6. Rapamycin and Dynasore Do Not Alter the RBD-Induced EC Permeability and vWF Secretion

Since the mammalian target of rapamycin (mTOR) is modulated in SARS-CoV-2 infected cells [55], rapamycin, an mTOR inhibitor, can be repurposed at low dosages for the treatment of COVID-19 [56]. Dynasore, an inhibitor of dynamin, reduces the endocytosis of the SARS-CoV-2 spike protein, suggesting that SARS-CoV-2 enters the cells through endocytosis [57]. Based on the role of mTOR and endocytosis in SARS-CoV-2 infection, we hypothesized that rapamycin and dynasore may affect SARS-CoV-2 spike protein-induced EC permeability and vWF secretion in EA.hy 296 cells. To test this hypothesis, we analyzed the effect of rapamycin or dynasore pretreatment on the EC permeability and vWF secretion in SARS-CoV-2 spike S1 RBD-untreated or -treated cells for 72 h. However, both rapamycin and dynasore did not inhibit the RBD-induced EC permeability (Figure 7a). Instead, they induced the EC permeability in the RBD-untreated cells. Similarly, we did not see any inhibition in the RBD-induced vWF secretion from the cells by rapamycin and dynasore (Figure 7b). However, these inhibitors have no effect on vWF secretion from the RBD-untreated cells. These findings suggested that rapamycin and dynasore may have non-specific effects on EC permeability.

## 3. Discussion

It has been suggested that the SARS-CoV-2 spike protein causes not only a leaky vascular barrier but also increases the expression of vWF [22]. Thus, several clinical studies suggested that vWF is the potential biomarker of endothelial damage and thrombotic risk in COVID-19 as extremely high levels of vWF are common in COVID-19 patients [58,59]. In this study, we analyzed in vitro the SARS-CoV-2 spike protein inducing the EC permeability and vWF secretion and dissected the signaling pathway required for it. We first verified, through the assessment of the dose- or time-dependent effect, that the SARS-CoV-2 spike protein S1 RBD is sufficient to induce the EC permeability and vWF secretion. We also assessed the proinflammatory cytokine (TNF-α, IL-6, and IL-1β) levels in RBD-treated cells as COVID-19 leads to a cytokine storm [7]. However, we only saw a negligible increase (TNF-α) or no increase in secretion (IL-6 and IL-1β) from cells treated with the RBD from 24–120 h. We then determined the effect of various mutations (including those in SA and SC variants) in SARS-CoV-2 spike protein S1 on the EC permeability and vWF secretion. Our results suggested that the mutants we tested in this study behave like wild-type spike protein S1 in inducing the EC permeability and vWF secretion. 

Since ACE2 mediates SARS-CoV-2 infection [60], we then assessed the expression of ACE2 in the RBD-treated cells and found that ACE2 expression levels were slightly elevated in the RBD-treated cells. However, a previous study [28] showed a slight reduction in ACE2 levels in pulmonary arterial ECs when treated with the spike protein. Although the exact reason for this variation is unknown, it could be due to the difference in the spike protein treatment time and/or the cells used for the ACE2 expression analysis. A recent study identified differences, particularly in endothelial permeability, between EA.hy296 cells and primary human aortic ECs [61]. Although the use of EA.hy296 cells is a limitation of this study, most of the results we obtained for the S1 protein-induced EC permeability using EA.hy296 cells matched that obtained by using primary microvascular ECs (MVECs) by others [28,29]. Since COVID-19 is microvascular endotheliopathy, it would be interesting to confirm our findings using primary MVECs in future studies. Since the SARS-CoV-2 nucleocapsid (N) protein circulates, like the spike protein, in COVID-19 patients and would not bind to the ACE2 receptor, it would be a good control to be included in future studies on the spike protein-induced EC permeability [30].

The excess vascular leak seen in bacterial LPS-triggered sepsis is caused by the activation of ARF6 [62]. COVID-19 causes leaky vasculature like in LPS-induced sepsis [12] and, therefore, we assessed whether ARF6 activation is altered in the RBD-treated cells. Our results suggest that the activation of ARF6, but not functionally distinct ARF1, is increased in a dose-dependent manner in the RBD-treated cells. We used the ACE2 chemical inhibitor dalbavancin and the inhibitory antibody to analyze the role of ACE2 in the RBD-induced EC permeability and vWF secretion. Both inhibitors effectively inhibited the RBD-induced EC permeability. Chlortetracycline (the ARF6 activation inhibitor), but not QS11 (the inhibitor of ARF6 inactivation) also inhibited the EC permeability and vWF secretion, indicating that ARF6 activated in the RBD-treated cells play a role in the RBD-induced EC permeability and vWF secretion. A previous study has shown that the SARS-CoV-2 spike protein causes endothelial dysfunction by binding to ACE2 and thereby increasing redox stress through impairing mitochondrial function [28]. In addition to this, we have shown in this study that NAC, a chemical inhibitor that reduces oxidative stress, inhibits RBD-induced EC permeability and vWF secretion. 

Since NF-κB, Mtor, and dynamin play a role in SARS-CoV-2 infection and ROCK1/2 activation is important for thrombin-induced EC permeability, we also analyzed the effect of the inhibitors of these signaling molecules in the RBD-induced EC permeability and vWF-secretion. SC514 (NF-κB) and Y27632 (ROCK1/2 inhibitor) significantly inhibited the RBD-induced EC permeability and vWF secretion. However, rapamycin and dynasore failed to reduce EC permeability and vWF secretion in the RBD-treated cells and even induced high EC permeability in the RBD-untreated cells, suggesting they may be toxic to the cells. Based on the findings of our study and that of others, we proposed a signaling cascade downstream of ACE2 for SARS-CoV-2 spike protein-induced EC permeability and its vWF secretion (Figure 8). The SARS-CoV-2 spike protein S1 binds to ACE2 to activate ARF6 and the NF-κB signaling pathway, which may cause mitochondrial dysfunction as a result of oxidative stress promoted by ROS induction, increasing MLC-p mediated by ROCK, leading to reorganization of the actin cytoskeleton into contractile stress fibers. This results in the opening of intercellular gaps, ultimately causing increased EC permeability and vWF secretion. Since the spike protein and cryptic SARS-CoV-2 tissue reservoirs have been suggested to be associated with the endothelial injury of long COVID [63], it is tempting to speculate that vascular permeability may mediate some of the effects of long COVID. When SARS-CoV-2 infection leads to vascular permeability, which ultimately results in the development of multi-organ tissue injury in long COVID [64].

## 4. Materials and Methods

### 4.1. Materials

All chemicals used in this study were obtained from Sigma-Aldrich unless otherwise specified. SC514 was purchased from Tocris. Rapamycin was obtained from LC Laboratories. Y27632 dihydrochloride was purchased from Hello Bio. Dynasore was obtained from Ascent Scientific. SARS-CoV-2 (2019-nCoV) spike protein S1 RBD (R319-F541, His-tagged) and vWF ELISA Kit were purchased from Sino Biological. SARS-CoV-2 spike protein S1 mutant sampler set containing wild-type (WT), D614G mutant, N439K mutant, SA variant (K417N/E484 K/N501Y), and SC variant (S13I/W152C/L452R) was obtained from Abenomics. Cy3-conjugated anti-mouse immunoglobulin (Ig)G was purchased from Jackson ImmunoResearch. The anti-ARF1 antibody, rabbit polyclonal, was described previously [39]. The ARF6(3A-1) mouse monoclonal antibody and the normal goat antibody were obtained from Santa Cruz Biotech. The ACE2 inhibitory antibody (goat polyclonal) and human TNF-α, IL-6, and IL-1β DuoSet ELISA kits were purchased from R&D Systems. 96-well µClear^®^ half-area black plates with flat bottoms were obtained from Greiner Bio-one. Enhanced chemiluminescence (ECL) Select immunoblotting Detection Reagent was from Cytiva. Restore plus stripping buffer and ultra TMB substrate were obtained from Thermo Fisher Scientific. ChemiDoc XRS imaging system was from Bio-Rad.

### 4.2. Cell Culture

Endothelial cell line EA.hy 296 (#CRL-2922, ATCC) was cultured in Dulbecco’s Modified Eagles Medium (DMEM) with high glucose and sodium pyruvate supplemented with 0.1 mM hypoxanthine, 0.4 µM aminopterin, 16 µM thymidine, 10% fetal bovine serum (FBS), 2 mM L-glutamine, 100 U/mL penicillin, and 0.1 mg/mL streptomycin (full serum medium [FSM]) in a humidified incubator at 37 °C and 5% CO_2_.

### 4.3. 96-Well Trans-Monolayer Endothelial Permeability Assay

This was carried out as described previously with some modifications [40]. Briefly, the wells of a 96-well µClear^®^ half-area black plate with flat bottom were coated with biotin-gelatin by adding 25 μL of 0.25 mg/mL biotin-gelatin made in phosphate-buffered saline (PBS) in each well and incubating the plate for 60 min at room temperature. Thereafter, 80,000 EA.hy 296 cells were seeded into each of the biotin-gelatin coated wells and grew at 5% CO_2_/37 °C for 4 days to reach 100% confluency (and thereby became a monolayer). The cells were pre-incubated for 30–60 min without or with specific inhibitors and then stimulated without or with the SARS-CoV-2 spike protein S1 or S1 RBD in phenol red-free DMEM with 0.2% fat-free BSA (PRF medium) for the indicated time. After the treatment, the medium was removed from each well and stored at 20 °C for analysis of the secreted vWF, and the cytokines (TNF-α, IL-6, and IL-1β) by ELISA. Then, the cells were incubated with FITC-streptavidin (25 μg/mL) in PRF medium for 10 min and fixed with 4% paraformaldehyde (PFA) for 20 min. The cells were then washed with PBS and left in Prolong Live antifade reagent (diluted 1 in 100 FluoroBrite DMEM) at 4 °C or used immediately to visualize by fluorescence microscopy at 10× magnification and assessed the percentage permeability by quantifying the fluorescence intensity by ImageJ.

### 4.4. ARF Activation Assay

This assay was carried out as described previously with some modifications [65]. Briefly, EA.hy 296 cells grown to 100% confluence in 6 cm plates were treated with 0–10 μg/mL SARS-CoV-2 spike S1 RBD protein for 3 days. After the incubation, the cells were washed twice with 2 mL of cold PBS and lysed using 500 μL of pulldown lysis buffer (25 mM Tris-HCl, pH 7.2, 150 mM NaCl, 5 mM MgCl_2_, 1% NP40, 5% glycerol) containing 1% mammalian protease inhibitor mix, 10 mM sodium fluoride, and 1 mM sodium orthovanadate. Thereafter, 350 μL of the lysate incubated with 20 μL glutathione s-transferase-Golgi-associated, adaptin ear-containing, ARF-binding protein (GGA)3 protein-binding domain beads (25% resin containing 5 mg protein/mL of resin) at 4 °C with continual mixing. After 2 h incubation, the beads were washed three times with pulldown lysis buffer, resuspended in 1× sodium dodecyl sulfate-polyacrylamide gel electrophoresis (SDS-PAGE) sample buffer, boiled at 100 °C for 5 min, and stored at −20 °C. For the input controls, 100 μL of the lysate was mixed with 25 μL 5× SDS-PAGE sample buffer (125 mM Tris HCl, pH 6.8, containing 5% SDS, 50% glycerol, 0.005% bromophenol blue, and 5% β-mercaptoethanol) and boiled at 100 °C for 5 min. 

### 4.5. Immunoblotting

EA.hy 296 cells were lysed by using TRI Reagent and carried out immunoblotting as described previously [66,67]. The protein content of the lysates was estimated using a bicinchoninic acid (BCA) protein kit and following the manufacturer’s instructions. The lysates were mixed with 0.25 volume of 5X SDS-PAGE sample buffer, heated at 100 °C for 5 min, and stored at −20 °C. The protein samples resolved by SDS-PAGE were transferred onto a polyvinylidene difluoride (PVDF) membrane. All the following incubations were carried out at room temperature unless otherwise specified. The PVDF membrane was blocked using 5% non-fat milk made in Tris-buffered saline containing 0.1% Tween (TBST) (blocking buffer) for an hour. The blot was probed with the primary antibody diluted in a blocking buffer overnight at 4 °C. The membrane was then washed three times with TBST and incubated for 1 h with the HRP-conjugated secondary antibody, which diluted 1 in 2500 in blocking buffer. It was then washed in TBST and developed by incubating in ECL Select immunoblotting Detection Reagent. The bands on the developed blot were visualized using the ChemiDoc XRS imaging system. For stripping and reprobing, the membranes were incubated with Restore plus stripping buffer for 15 min. After washing the membrane with TBST, the probing process was carried out by starting at the blocking step as described above. The intensity of bands in immunoblots was quantified by densitometric analysis and normalized to the expression of loading control (β-actin) or total ARF.

### 4.6. Quantification of Secreted vWF and Cytokines by ELISA

By using ELISA Kits, vWF and three important proinflammatory cytokines (TNF-α, IL-6, and IL-1β) secreted from EA.hy 296 cells were measured. The ELISA assays were carried out at room temperature, unless otherwise specified, by following the manufacturers’ instructions. Briefly, a 96-well microplate was coated with capture antibody by incubating the plate overnight at 4 °C. After washing with wash buffer (0.05% Tween20 in TBS, pH 7.4), the wells were blocked for 1 h using blocking buffer (2% BSA made in wash buffer). The wells were then incubated with samples or standards diluted in sample dilution buffer (0.1% BSA in wash buffer) for 2 h. After washing with wash buffer, the wells were incubated with the biotin-conjugated detection antibody diluted in sample dilution buffer for 2 h. Again, the wells were washed with wash buffer and incubated with HRP-conjugated Streptavidin for 20 min. The wells were then washed with wash buffer and incubated with Ultra TMB substrate for 20 min in dark. After stopping the reaction by adding 2 M H_2_SO_4_, its optical density at 450 nm (OD_450_) was measured using a 96-well plate reader.

### 4.7. Statistics

Image J software was used for densitometric analysis. Data were analyzed using GraphPad Prism 6 software. For all data, comparisons were made using Ordinary one/two-way analysis of variance (ANOVA). Results are shown as the mean ± SEM with *p* < 0.05 considered statistically significant.

## 5. Conclusions

We have shown in this study that the SARS-CoV-2 spike protein S1 RBD significantly induces leaky endothelial vascular permeability and vWF secretion, which are unaffected by the mutations in the spike protein including those found in SA and SC variants. We showed alteration in ACE2 expression and ARF6 activation in the RBD-treated cells. The inhibitors of ACE2, ARF6 activation, NF-κB activation, and ROS and ROCK significantly inhibited SARS-CoV-2 spike protein S1 RBD-induced EC permeability and vWF secretion. We proposed a signal cascade for RBD-induced EC permeability and vWF secretion. Further investigation is needed to find missing steps in the signaling pathway for RBD-induced permeability and vWF secretion. This will yield potential targets for developing novel drugs or repurposing existing drugs to treat SARS-CoV-2 strains that evade immunity by COVID-19 vaccines.

## Figures and Tables

**Figure 1 ijms-24-05664-f001:**
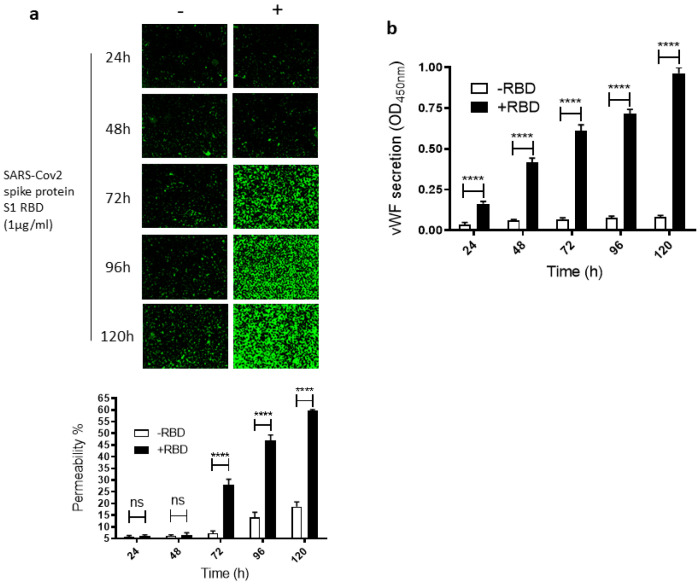
Time-dependent effect of the SARS-CoV-2 spike protein S1 RBD on EA.hy 296 cell monolayer permeability and its vWF secretion. EA.hy 296 cells grown on biotinylated gelatin to confluence were treated without or with 1 µg/mL SARS-CoV-2 spike protein S1 RBD for 24–120 h. After the incubation, the medium was removed (to use for analyzing the secretion of vWF and the proinflammatory cytokines), and FITC-streptavidin was added to the cells for 10 min. Then, the cells were washed and fixed with 4% PFA, and FITC-streptavidin bound to biotinylated gelatin was visualized by fluorescence microscopy and quantified fluorescence intensity by using Image J software. Green fluorescence indicates areas of the EA.hy 296 cell monolayer permeable to FITC-conjugated streptavidin. Images were taken at 10× magnification using a fluorescence microscope. (**a**) Visualization of time-dependent induction of permeability in EA.hy 296 cell monolayers by 1 µg/mL SARS-CoV-2 spike S1 RBD treatment. (**lower panel**) Represents the quantification data of (**a**), which was presented as the percentage (%) permeability of the EA.hy 296 cell monolayer. (**b**) vWF secretion by EA.hy 296 cell monolayer treated without (−RBD) or with 1 µg/mL RBD (+RBD) for 24–120 h. Statistical significance was measured by two-way ANOVA by comparing the treated with the untreated control (−RBD) at each time point (**** *p*  <  0.0001. ns, not significant).

**Figure 2 ijms-24-05664-f002:**
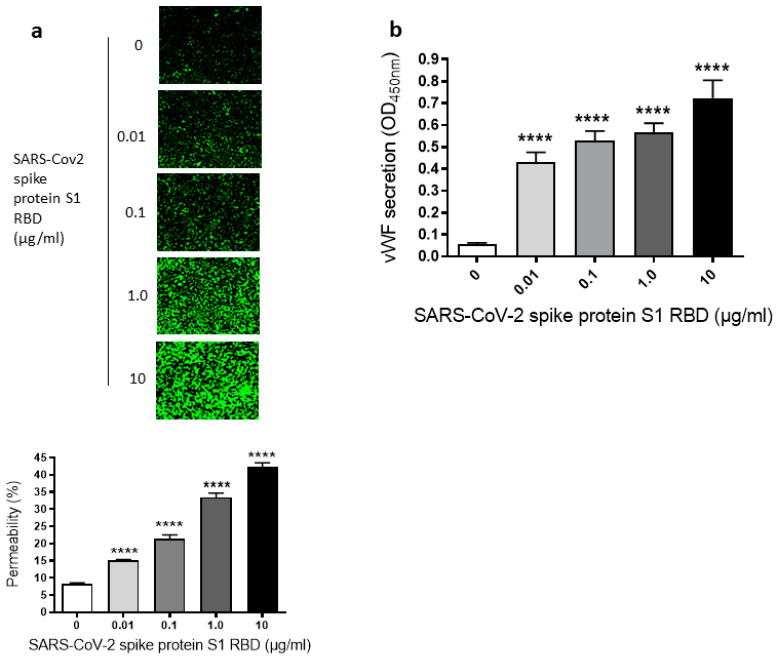
Dose-dependent effect of SARS-CoV-2 spike S1 RBD on EA.hy 296 cell monolayer permeability and vWF secretion. (**a**) Visualization at 10× magnification of 0–10 µg/mL SARS-CoV-2 spike S1 RBD-induced permeability in EA.hy 296 cell monolayer. (**lower panel**) Represents the quantification data of (**a**). (**b**) vWF secretion by EA.hy 296 cell monolayers treated with 0–10 µg/mL RBD for 72 h. Statistical significance was measured by one-way ANOVA comparing the treated with the untreated control (**** *p*  <  0.0001).

**Figure 3 ijms-24-05664-f003:**
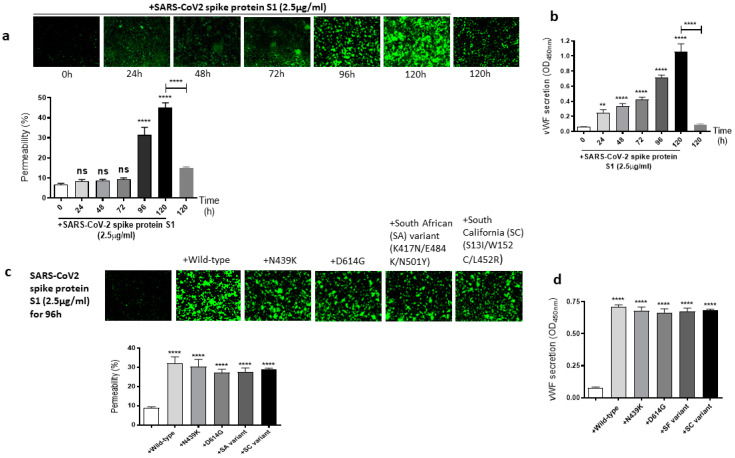
Effect of the SARS-CoV-2 spike protein S1 and its mutants on EA.hy 296 cell monolayer permeability and its vWF secretion. (**a**) Visualization at 10× magnification of SARS-CoV-2 spike protein S1-induced permeability in a time-dependent manner in EA.hy 296 cells monolayers. (**lower panel**) Represents the quantification data of (**a**). (**b**) vWF secretion by EA.hy 296 cell monolayer treated with 2.5 µg/mL spike protein S1 for 0–120 h. (**c**) Visualization at 10× magnification of permeability of EA.hy 296 cell monolayer treated without or with 2.5 µg/mL of the SARS-CoV-2 spike protein S1 (WT) or its mutants for 96 h. (**lower panel**) Represents the quantification data of (**c**). (**d**) vWF secretion by EA.hy 296 cell monolayers treated with 2.5 μg/mL spike protein S1 or its mutants for 96 h. Statistical significance was measured by two-way ANOVA comparing the treated with the untreated control (** *p*  <  0.01, **** *p*  <  0.0001. ns, not significant).

**Figure 4 ijms-24-05664-f004:**
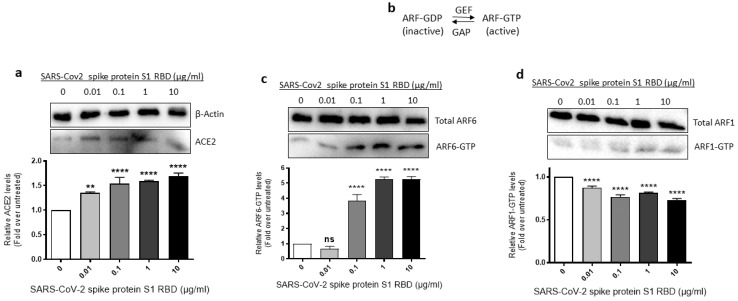
Effect of the SARS-CoV-2 spike protein S1 RBD treatment of EA.hy 296 Cells on ACE2 expression and ARF6 activation. EA.hy 296 cells were treated with 0–10 μg/mL RBD manner for 72 h and lysed. (**a**) The cell lysates were analyzed for ACE2 protein expression by immunoblotting using the ACE2 inhibitory antibody or β-actin antibody. (**lower panel**) The intensity of the bands in (**a**) was quantified and normalized to the expression of loading control (β-actin). (**b**) Schematic of the inactivation and activation cycle of ARF and its regulation by GEFs and GAPs. The cell lysates were subjected to GST-GGA3 PBD pulldown to quantify ARF6-GTP (**c**) or ARF1-GTP (**d**). ARF6-GTP or ARF1-GTP levels in the pulldown and total ARF6 or ARF1 levels were assessed by immunoblotting using the ARF6 or ARF1 antibody. The intensity of the bands in the ARF6-GTP (**c**, **lower panel**) or ARF1-GTP (**d**, **lower panel**) blot was quantified and normalized to the expression of total ARF6 or ARF1 present in the corresponding sample. Statistical significance was measured by one-way ANOVA comparing the treated with the untreated control (** *p*  <  0.01, **** *p*  <  0.0001. ns, not significant).

**Figure 5 ijms-24-05664-f005:**
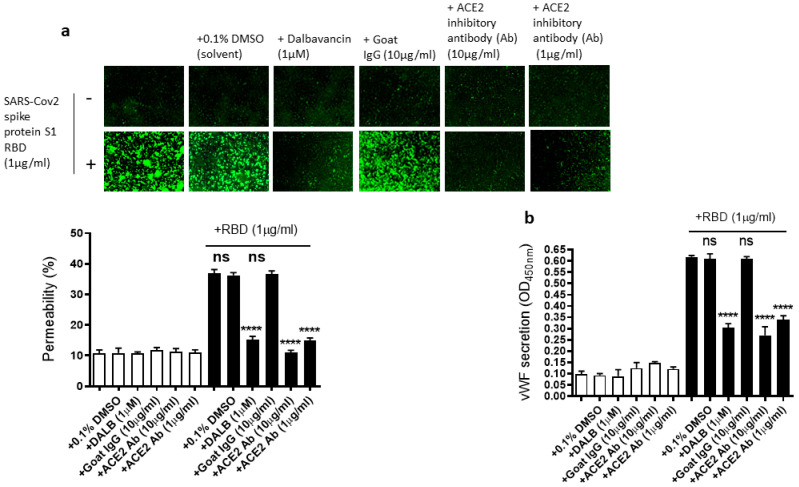
Effect of dalbavancin and the ACE2 inhibitory antibody on SARS-CoV-2 spike protein S1 RBD-induced EA.hy 296 cell monolayer permeability and its vWF secretion. EA.hy 296 cell monolayer pre-incubated without or with DMSO (0.1%), dalbavancin (1 μM), or goat IgG (10 μg/mL), or ACE2 inhibitory goat IgG (1 or 10 μg/mL) for 30–60 min were treated without or with 1 μg/mL SARS-CoV-2 spike S1 RBD for 72 h. (**a**) Visualization at 10× magnification of the RBD-induced EC permeability inhibition by dalbavancin and the ACE2 inhibitory antibody. (**lower panel**) Represents the quantification data of (**a**). (**b**) Inhibition of the RBD-induced vWF secretion from EA.hy 296 cells by dalbavancin or the ACE2 inhibitory antibody. Statistical significance was measured by one-way ANOVA comparing the treated group with the untreated control group (**** *p*  <  0.0001. ns, not significant).

**Figure 6 ijms-24-05664-f006:**
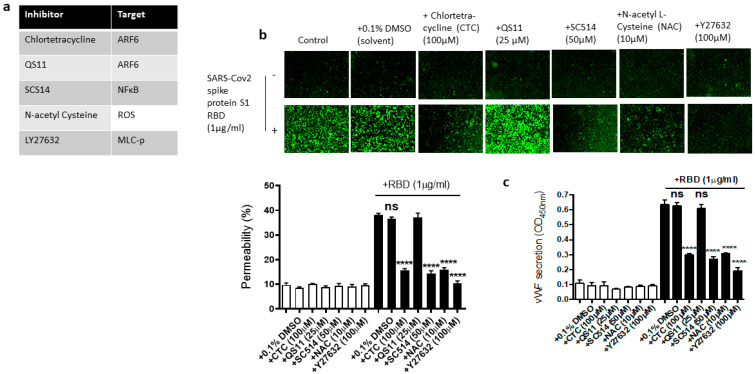
Effect of various inhibitors of ACE2 downstream signaling pathway on the EC permeability and vWF in RBD-stimulated EA.hy 296 cells. The cells pre-incubated for 1–2 h without or with the indicated inhibitor were treated without or with 1 μg/mL of the SARS-CoV-2 spike protein S1 RBD for 72 h. (**a**) The list of inhibitors (and their targets) used in this study. (**b**) Visualization at 10× magnification of the RBD-induced EC permeability inhibition by the indicated inhibitor. (**lower panel**) Represents the quantification data of (**b**). (**c**) Inhibition of the RBD-induced vWF secretion from EA.hy 296 cells by the indicated inhibitor. Statistical significance was measured by one-way ANOVA comparing the RBD-treated group with the control RBD-untreated group (**** *p*  <  0.0001. ns, not significant).

**Figure 7 ijms-24-05664-f007:**
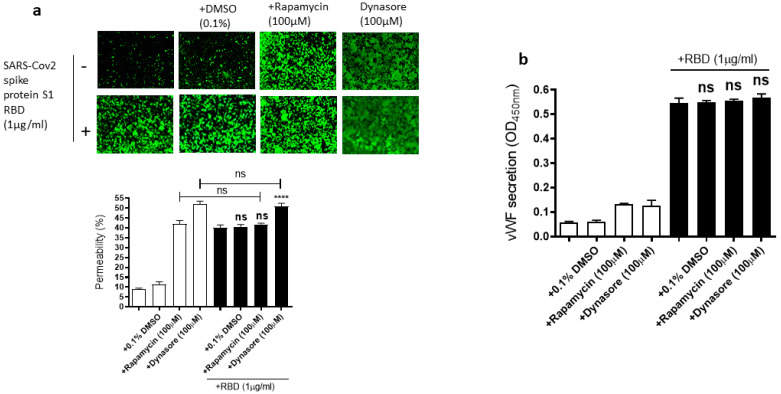
Effect of rapamycin and dynasore on the RBD-induced EC permeability and vWF secretion. EA.hy 296 cells pre-incubated without or with rapamycin or dynasore for 1 h were treated without or with 1 μg/mL SARS-CoV-2 the spike protein S1 RBD for 72 h. (**a**) Visualization at 10× magnification of the effect of rapamycin and dynasore on the RBD-induced EC permeability. (**lower panel**) Represents the quantification data of (**a**). (**b**) The effect of rapamycin and dynasore on the RBD-induced vWF secretion from EA.hy 296 cells. Statistical significance was measured by one-way ANOVA comparing the RBD-treated group with the control RBD-untreated group (**** *p*  <  0.0001. ns, not significant).

**Figure 8 ijms-24-05664-f008:**
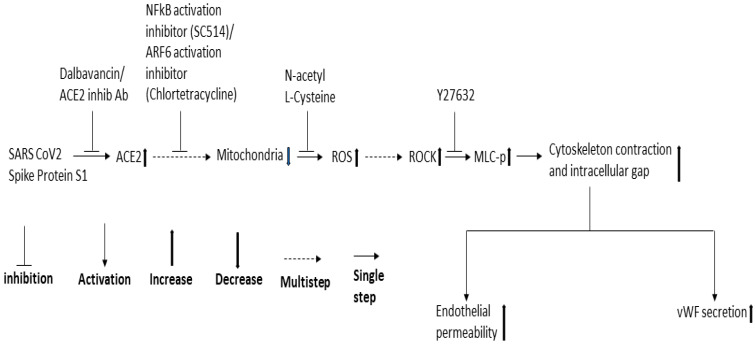
Proposed schematic representation of the signaling cascade downstream of ACE2 for SARS-CoV-2 spike protein-induced EC permeability and its vWF secretion.

## Data Availability

Not applicable.

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
