# Peer review of "Molecular Analysis of SARS-CoV-2 Spike Protein-Induced Endothelial Cell Permeability and vWF Secretion"

_ijms, 2023, doi:10.3390/ijms24065664_

Round 1

Reviewer 1 Report

Review for International Journal of Molecular Sciences

The title of this review is “Molecular analysis of SARS-CoV-2 spike protein-induced endothelial cell permeability and vWF secretion”. Herein, the authors show spike protein increased the ACE2 that ultimately induced the endothelial cell permeability and vWF secretion. Overall, the data provided by the authors were redundant and not well organized. In addition, the data shown here is controversial with the previous reports. However, providing data is not enough to peruse. Here are several points that should help strengthen the manuscript and improve its clarity

1.    Figure 1> Please change ai to A, aii to A. aii represents the quantification data of ai. The denotation of ai and aii is unusual. In addition, in both 24h and 48h, FITC signal is stronger in the sample without S1 RBD than with S1 RBD (it also shows more cells). Please revisit this issue and also please provide phase-contrast figure of both.

2.    Figure 3 ai and aii> The authors just display similar data which authors show in figure 1 and figure 2. It did not offer new information. In addition, in figure cii, What is the white stick indicate?

3.    Figure 4. Need to explain how spike s1 RBD increases ACE2 gene expression? If this is true, what about other spike mutants which have different binding affinities to ACE2? Lui et al., (Circulation Research 128(9) 2021) already show that Spike protein can damage vascular endothelial cells by downregulating ACE2. The authors results are controversial. Why? The authors continuously performed permeability and vWF secretion assay. They should include several different technics to peruse their hypothesis. 

Author Response

We would like to thank the reviewer for sparing his/her valuable time in reading the manuscript and providing comments. We have responded to the specific comments raised by the reviewer in the following way. Also, we included the missing Figure 8 (Proposed schematic representation of the signalling cascade downstream of ACE2 for SARS-CoV-2 spike protein-induced EC permeability and its vWF secretion) at the end of the discussion section in the revised manuscript.

RESPONSE TO REVIEWER #1 COMMENTS

  1. Figure 1> Please change ai to A, aii to A. aii represents the quantification data of ai. The denotation of ai and aii is unusual. In addition, in both 24h and 48h, FITC signal is stronger in the sample without S1 RBD than with S1 RBD (it also shows more cells). Please revisit this issue and also please provide phase-contrast figure of both.

As kindly suggested by this reviewer, in Figure 1 (also in other Figures where we mentioned ai and aii or ci and cii), changed (ai) to (a) and (aii) to (lower panel), corrected the figure legend as well as the text to reflect this. After treating the cells with the S1 RBD for 24h-120h, cells were fixed and stained with DAPI to confirm that the increase in FITC signal is not due to the loss of cells upon spike protein treatment. Since we didn’t see any cell death, we haven’t done DAPI staining of the cells when we repeated time-course and the dose-dependent effect of the S1 RBD on endothelial permeability. So, we replaced Figures 1a and 2a with the samples for which we did DAPI staining in the revised manuscript (see Figure S3; lines 112-115 and 151-153). Please note that the FITC signal appears to be slightly increased in the sample untreated for 24h or 48h (in the old Figure 1a), which could be an experimental variability since we didn’t see that in the sample untreated for 24h or 48h samples in the revised Figure 1a.

  1. Figure 3 ai and aii> The authors just display similar data which authors show in figure 1 and figure 2. It did not offer new information. In addition, in figure cii, What is the white stick indicate?

Please note that we used the S1 RBD in Figures 1 and 2 whereas the spike protein S1 in Figure 3. Since the mutants were made in the S1 protein, we analysed the time-dependent effect of the spike protein S1 on the EC permeability and vWF secretion. So, the data in Figures 1 and 3 are not the same. The white unfilled column in that figure indicates cells untreated for 96h (corrected Figure 3a with proper labelling in the revised manuscript and changed the corresponding column colour in the graph [lower panel]).

  1. Figure 4. Need to explain how spike s1 RBD increases ACE2 gene expression? If this is true, what about other spike mutants which have different binding affinities to ACE2? Lui et al., (Circulation Research 128(9) 2021) already show that Spike protein can damage vascular endothelial cells by downregulating ACE2. The authors results are controversial. Why? The authors continuously performed permeability and vWF secretion assay. They should include several different technics to peruse their hypothesis.

We have seen a slight increase in ACE2 expression in the S1 RBD-treated cells whereas Lui et al. (Ref 28) have seen a slight decrease in ACE2 expression in the cells exposed to spike protein-expressing virus. We explained this variability in the discussion section (see lines 354-361 in the revised manuscript). We used not only permeability and vWF secretion assays but also the ARF6 activation and cytokine secretion assays.

Reviewer 2 Report

The manuscript "Molecular analysis of SARS-Cov-2 spike protein-induced endothelial cell permeability and vWF secretion" addresses an important issue: The molecular analysis of SARS-Cov-2 spike protein-induced endothelial cell permeability and vWF secretion. This comprehensive and multi-faceted study reveals potential targets for developing novel drugs or repurposing existing drugs to treat SARS-CoV-2 strains that evade immunity by COVID-19 vaccines.

The objectives were clearly stated and explained in the manuscript, and the experimental strategy was appropriate to gather the experimental information from which the conclusions were drawn. The manuscript is well written and has good organization. The authors have done a great job on analyzing the experimental data and on discussing the results, considering always different alternative explanations/considerations for interpreting the results.

Due to the high impact and scientific soundness of the findings covered in this article, as well as, the high quality of the research performed I would suggest to accept the paper in present form.

Author Response

We would like to thank the reviewer for sparing his/her valuable time in reading the manuscript and providing comments. There are no specific comments from this reviewer that need our response. We included the missing Figure 8 (Proposed schematic representation of the signalling cascade downstream of ACE2 for SARS-CoV-2 spike protein-induced EC permeability and its vWF secretion) at the end of the discussion section in the revised manuscript.

Reviewer 3 Report

  1. P. 2, 1st and 2nd paragraphs: A “cytokine storm” is part of severe COVID-19. However, similar elevated levels of these cytokines, or even higher levels, are found in non-COVID types of ARDS even though the latter lack the microvascular EC injury and thrombosis found throughout the body and characteristic of severe SARS-CoV-2 infection. The authors should note that the main cause of mortality in COVID-19 is a microvascular thrombosis (e.g., reviewed in Chen W, Biological Procedures 2021;23:4.
  2. As a corollary of point no. 1, in addition to ref. 11 and 12 the authors should include the original references documenting that COVID-19 is a systemic microvascular thrombotic disease (Magro C, Transl Res 2020;220:1 and Ackerman M, N Engl J Med 2020;383:120).
  3.  Why did most experiments shown after the dose-response in Fig. 1b use 1ug/ml of S1 protein? Fig. 1B shows virtually no dose-response between 0.01 and 1ug/ml, at least in terms of vWF release. It was only between 1 and 10ug/ml that any real difference is seen.
  4.  Were any protein controls used? Since this manuscript bases its conclusions on S1 binding to ACE receptors, a good control would be to test SARS-CoV-2 N (nucleocapsid) protein. Like S, N protein also circulates in SARS-CoV-2 individuals but would not interact with that receptor. It is readily available commercially.
  5. Were the S1 proteins used tested for endotoxin content?
  6. I am most concerned about the model EC cell line used for all experiments. EA.hy296 is a very artificial construct, a somatic cell hybrid between a human large vessel EC (HUVEC) and a human lung epithelial cell. As noted above, COVID-19 is a microvascular endotheliopathy; it does not directly affect large vessel EC to an appreciable extent and most in vitro models I’ve seen published using SARS-CoV-2 proteins involve primary microvascular EC. The best confirmation that the EA.hy296 model is of any in vivo relevance to S protein-induced permeability validity would be to perform at least one experiment replicating those results with primary human MVEC. This is particularly important for the hypothesis in this manuscript. Specifically, a recent review of in vitro EC models for cardiovascular disease contrasted EA.hy296 with primary human aortic EC. Major differences were documented, most strikingly in the one pathway—vascular permeability—that the authors are examining. (Wang D, Current Res Biotechnol 2021;3:135.) This should at least be noted as a limitation.
  7. S protein and cryptic SARS-CoV-2 reservoirs have been suggested as involved in the endothelial injury of long COVID (e.g., Ahamed J, J Clin Invest 2022;132:e161167). The authors might speculate as to the role of vascular permeability in mediating some of the effects of long COVID.  

Author Response

We would like to thank the reviewer for sparing his/her valuable time in reading the manuscript and providing comments. We have responded to the specific comments raised by the reviewer in the following way. Also, we included the missing Figure 8 (Proposed schematic representation of the signalling cascade downstream of ACE2 for SARS-CoV-2 spike protein-induced EC permeability and its vWF secretion) at the end of the discussion section in the revised manuscript.

RESPONSE TO REVIEWER #3 COMMENTS

  1. P. 2, 1 and 2 paragraphs: A “cytokine storm” is part of severe COVID-19. However, similar elevated levels of these cytokines, or even higher levels, are found in non-COVID types of ARDS even though the latter lack the microvascular EC injury and thrombosis found throughout the body and characteristic of severe SARS-CoV-2 infection. The authors should note that the main cause of mortality in COVID-19 is a microvascular thrombosis (e.g., reviewed in Chen W, Biological Procedures 2021;23:4.).

We would like to thank the reviewer for providing clarification and suggesting including a reference. We agree with the suggestion. By considering the reviewer’s suggestions, we included the suggested reference (Ref 27) and added the following in the revised manuscript (see lines 70-71).

“Further, the main cause of mortality in COVID-19 patients is microvascular thrombosis (Ref 26).”

  1. As a corollary of point no. 1, in addition to ref. 11 and 12 the authors should include the original references documenting that COVID-19 is a systemic microvascular thrombotic disease (Magro C, Transl Res 2020;220:1 and Ackerman M, N Engl J Med 2020;383:120).

We would like to thank the reviewer for the suggestion. We added suggested references in the revised manuscript (see lines 63-64 [Ref 13 & 14]).

  1. Why did most experiments shown after the dose-response in Fig. 1b use 1ug/ml of S1 protein? Fig. 1B shows virtually no dose-response between 0.01 and 1ug/ml, at least in terms of vWF release. It was only between 1 and 10ug/ml that any real difference is seen.

We used 1µg/ml concentration (sub-optimal concentration) in the permeability assay since it not only produced a clear response but also would allow analysing of the positive or negative effects of the treatments on the RBD-induced EC permeability in further experiments. We measured vWF secretion in the medium in which the cells incubated with the S1 RBD and hence we used the same concentration for vWF release. As the reviewer kindly suggested, we saw some difference between 1µg/ml and 10µg/ml and therefore 1µg/ml of the RBD can also be considered as a sub-optimal concentration for vWF release.

  1. Were any protein controls used? Since this manuscript bases its conclusions on S1 binding to ACE receptors, a good control would be to test SARS-CoV-2 N (nucleocapsid) protein. Like S, N protein also circulates in SARS-CoV-2 individuals but would not interact with that receptor. It is readily available commercially.

Unfortunately, we didn’t use it. Also, we didn’t know whether it interacts with ACE2 or not when we started our studies in 2020. We agree with this reviewer that it would be a good control to use in future studies on the spike protein-induced endothelial cell (EC) permeability (see lines 366-368 in the revised manuscript).

  1. Were the S1 proteins used tested for endotoxin content?

We used in this study the S1 RBD with His-tag at the C-terminus obtained from Sino Biologicals, which was expressed and purified from HEK293 cells, and tested for endotoxin content (low endotoxin, <1EU per µg protein, [https://www.sinobiological.com/recombinant-proteins/2019-ncov-cov-spike-40592-v08h]). Also, we used in this study the S1 mutant sampler set obtained from Abeomics, which was expressed and purified from CHO-K1 cells (https://www.abeomics.com/sars-cov-2-spike-s1-mutant-sampler-set). Although it is not mentioned, they should also have low endotoxin since they are expressed in CHO-K1 cells.

  1. I am most concerned about the model EC cell line used for all experiments. EA.hy296 is a very artificial construct, a somatic cell hybrid between a human large vessel EC (HUVEC) and a human lung epithelial cell. As noted above, COVID-19 is a microvascular endotheliopathy; it does not directly affect large vessel EC to an appreciable extent and most in vitro models I’ve seen published using SARS-CoV-2 proteins involve primary microvascular EC. The best confirmation that the EA.hy296 model is of any in vivo relevance to S protein-induced permeability validity would be to perform at least one experiment replicating those results with primary human MVEC. This is particularly important for the hypothesis in this manuscript. Specifically, a recent review of in vitro EC models for cardiovascular disease contrasted EA.hy296 with primary human aortic EC. Major differences were documented, most strikingly in the one pathway—vascular permeability—that the authors are examining. (Wang D, Current Res Biotechnol 2021;3:135.). This should at least be noted as a limitation.

We used EA.hy296 cells since they have been used by many groups in the past for EC permeability assay. Moreover, by using HUVEC cells, Rauti et al. (Ref 29 in the revised manuscript) have shown recently that most of the proteins encoded by the SARS-CoV-2 genome alter the EC permeability. Therefore, we and others used EA.hy296 as a cell line model to study the effect of the S1 protein on EC permeability. Further, most of the results we obtained for the S1 protein-induced EC permeability using EA.hy296 cells matched that obtained by using primary MVEC by others (see Ref. 28 and 29 in the revised manuscript). As the reviewer suggested, we mentioned the following in the revised manuscript (see lines 360-365).

“A recent study identified differences, particularly in endothelial permeability, between EA.hy296 cells and primary human aortic ECs. Although the use of EA.hy296 cells is a limitation of this study, most of the results we obtained for the S1 protein-induced EC permeability using EA.hy296 cells matched that obtained by using primary microvascular ECs (MVECs) by others. Since COVID-19 is microvascular endotheliopathy, it would be interesting to confirm our findings using primary MVECs in future studies.”

  1. S protein and cryptic SARS-CoV-2 reservoirs have been suggested as involved in the endothelial injury of long COVID (e.g., Ahamed J, J Clin Invest 2022;132:e161167). The authors might speculate as to the role of vascular permeability in mediating some of the effects of long COVID.

Please see lines 398-403 of the revised manuscript.

Round 2

Reviewer 3 Report

The authors have responding to my criticisms.